# New Prognostic Score (Essen Score) to Predict Postoperative Morbidity after Resection of Lung Metastases

**DOI:** 10.3390/cancers15174355

**Published:** 2023-09-01

**Authors:** Konstantinos Grapatsas, Fabian Dörr, Hruy Menghesha, Martin Schuler, Viktor Grünwald, Sebastian Bauer, Hartmut H. -J. Schmidt, Stephan Lang, Rainer Kimmig, Stefan Kasper, Natalie Baldes, Servet Bölükbas

**Affiliations:** 1Department of Thoracic Surgery, West German Cancer Center, Medical Faculty, University Hospital Essen, Ruhrlandklinik, Tueschner Weg 40, 45239 Essen, Germany; fabian.doerr@rlk.uk-essen.de (F.D.); hruy.menghesha@rlk.uk-essen.de (H.M.); natalie.baldes@rlk.uk-essen.de (N.B.); servet.boeluekbas@rlk.uk-essen.de (S.B.); 2Department of Medical Oncology, West German Cancer Center, Medical Faculty, University Hospital Essen, Hufelandstr. 55, 45147 Essen, Germany; martin.schuler@uk-essen.de (M.S.); stefan.kasper-virchow@uk-essen.de (S.K.); 3Interdisciplinary GU Oncology, West German Cancer Center Essen, Clinic for Urology and Clinic for Medical Oncology, University Hospital Essen, 45147 Essen, Germany; viktor.gruenwald@uk-essen.de; 4Department of Medical Oncology, Sarcoma Center, West German Cancer Center, Medical Faculty, University Hospital Essen, Hufelandstr. 55, 45147 Essen, Germany; sebastian.bauer@uk-essen.de; 5Department of Gastroenterology und Hepatology, Medical Faculty, University Hospital Essen, Hufelandstr. 55, 45147 Essen, Germany; hartmut.schmidt@uk-essen.de; 6Department of Otorhinolaryngology, Medical Faculty, University Hospital Essen, Hufelandstr. 55, 45147 Essen, Germany; stephan.lang@uk-essen.de; 7Department of Gynecology and Obstetrics, West-German Cancer Center, Medical Faculty, University Hospital Essen, Hufelandstr. 55, 45147 Essen, Germany; rainer.kimmig@uk-essen.de

**Keywords:** lung metastases, pulmonary metastasectomy, repeated metastasectomy, mortality, morbidity, morbidity score, thoracotomy, lobectomy

## Abstract

**Simple Summary:**

Pulmonary metastases from different primary tumors are a common issue in the everyday clinical praxis. The resection of lung metastases in patients selected by a multidisciplinary tumor board is a widely accepted surgical procedure that can prolong survival. The aim of the current retrospective study is to investigate postoperative morbidity and mortality after pulmonary metastasectomy, identify risk factors and develop a prognostic score to identify high-risk patients. We identified 1284 patients with lung metastases that were resected with curative intent. For increased postoperative morbidity, we identified cardiovascular comorbidities, major lung resections, repeated pulmonary metastasectomy and open thoracotomy. Based on these factors, the Essen score was developed. We believe that the Essen score is a useful tool to predict postoperative morbidity in patients undergoing pulmonary metastasectomy with curative intent.

**Abstract:**

Background: Pulmonary metastasectomy (PM) is a widely accepted surgical procedure. This study aims to investigate postoperative morbidity and mortality after PM and develop a score to predict high-risk patients. Methods: We retrospectively investigated all patients undergoing a PM in our institution from November 2012 to January 2023. Complications were defined as the diagnosis of any new disease after the PM up to 30 days after the operation. Results: 1284 patients were identified. At least one complication occurred in 145 patients (11.29%). Only one patient died during the hospital stay. Preoperative cardiovascular comorbidities (OR: 2.99, 95% CI: 1.412–3.744, *p* = 0.01), major lung resections (OR: 2.727, 95% CI: 1.678–4.431, *p* < 0.01), repeated pulmonary metastasectomy (OR: 1.759, 95% CI: 1.040–2.976, *p* = 0.03) and open thoracotomy (OR: 0.621, 95% CI: 0.415–0.930, *p* = 0.02) were identified as independent factors for postoperative complications. Based on the above independent factors for postoperative morbidity, the Essen score was developed (overall correct classification: 94.6%, ROC-Analysis: 0.828, 95% CI: 0.795–0.903). Conclusion: PM is a safe surgical procedure with acceptable morbidity and low mortality. The aim of the Essen score is to identify patients that are associated with risk for postoperative complications after PM.

## 1. Introduction

Lung metastases from different primary tumors are a very common issue in everyday clinical praxis. Nowadays, pulmonary metastasectomy (PM) is a widely accepted surgical procedure. PM with curative intent with complete resection of the lung metastases is associated with prolonged survival in highly selected patients [1]. PM is performed in a way to achieve lung-sparing resection of the lung metastases. Generally, PM is thought to be a safer surgical procedure in comparison with lung surgery for non-small cell lung cancer (NSCLC), in which major anatomical resections are usually needed [2]. Many studies investigate retrospective prognostic factors for survival after PM for different extrathoracic primary tumors. However, studies that evaluate risk factors for postoperative morbidity and mortality after PM are limited [2,3].

Our study aims to investigate postoperative morbidity and mortality after PM, identify risk factors and develop a predicting score for morbidity/mortality.

## 2. Materials and Methods

### 2.1. Study Design

Data were extracted from a prospectively maintained institutional database. We performed a retrospective single-center analysis of patients who underwent PM with curative intent at our department between November 2012 and January 2023. The study was designed and performed according to the revised Declaration of Helsinki. The study was approved by the university’s local ethics committee (Nr. 23-11273-BO).

### 2.2. Formation of the Study Population Exclusion/Inclusion Criteria

In the study were included patients who were deemed eligible for complete resection of pulmonary metastases within an oncological concept, approved by the institutional multidisciplinary tumor board. We divided the study population into two groups depending on the occurrence of postoperative complications.

Exclusion criteria were:–Patients < 18 years;–Diagnostic operations;–Metastasectomies with the intention of palliation.

### 2.3. Definition of Complications, Preoperative Comorbidities and Mortality

Postoperative complication or postoperative morbidity was defined the diagnosis of any new disease after the PM up to 30 days after the operation. An air leak through the chest tube >5 days after the operation was characterized as persistent. The complications were categorized according to the Clavien–Dindo classification as major and minor. In brief, grade I and II complications were characterized as minor. Grades III and IV complications were characterized as major. We further subdivided the complications to include major cardiopulmonary complications. As before, Grade III and IV cardiopulmonary complications were characterized as major. Complications leading to the death of the patient (Grade V) were not calculated separately, because of their low number [4].

Patients with age >70 years were characterized as elderly. Chronic obstructive pulmonary disease (COPD) was defined from the preoperative lung function [5]. Comorbidities that required pharmaceutical treatment were categorized as other comorbidities.

Mortality was defined as death by any cause up to 90 days after the operation. Mortality was divided into in-hospital mortality, 30-day mortality and 90-day mortality.

### 2.4. Surgical Procedure

For the resection of the pulmonary metastases with curative intent, the widely accepted criteria were followed. The primary tumor needed to be controlled or controllable. Patients with extrathoracic metastasis that was not resected or could not be resected were excluded. All of the pulmonary metastases must be resectable, with adequate pulmonary reserve after the PM. The operation was associated with acceptable low morbidity and mortality [5].

All patients were operated in general anesthesia. The operations were performed through open thoracotomy or with video-assisted thoracic surgery (VATS). In all operations, PM or repeated PM (RPM) was performed with curative intent. Anatomical resections such as lobectomies and pneumonectomies were characterized as major resections. Wedge resections and segmentectomies were characterized as minor resections.

### 2.5. Statistical Analysis

To compare different parameters, Fischer’s exact test and the Mann–Whitney U test were used when appropriate. Logistic regression models were used to select independent predictors for postoperative morbidity and mortality in our cohort. A *p*-value < 0.05 was considered statistically significant. Discrimination (ability of a scoring model to differentiate between complications and no complications) was evaluated with receiver operating characteristic (ROC) curves; the area under the curve (AUC) indicates the discriminative ability of the scores, i.e., the ability to discriminate complications from no complications. An AUC of 0.5 (a diagonal line) is equivalent to random chance, AUC > 0.7 indicates a moderate prognostic model, and AUC > 0.8 (a bulbous curve) indicates a good prognostic model. The overall correct classification (OCC) (the ratio of number of correctly predicted complications and no complications to the total number of patients) values of the scores were calculated. The risk of complications is given as odds ratios for all scores with 95% confidence intervals.

All statistical analyses were conducted using SPSS software (Version 27, IBM Corporation, New York, NY, USA).

## 3. Results

### 3.1. Study Population

We identified 1284 patients undergoing PM with curative intent for different primary tumors.

#### Surgical and Oncological Characteristics of the Study Population

Most of the patients had lung metastases from bone or soft tissue and other types of sarcoma (35.3%), followed by patients with colorectal cancer (14.9%). Most patients were operated through an open thoracotomy (56%). Most resections were lung-sparing wedge resections (84.5%) or segmentectomies (5.9%). In total, 9.2% of all resections were repeated pulmonary metastasectomies (RPM). Surgical and oncological characteristics are analyzed in Table 1.

### 3.2. Clinical Characteristics of the Study Population

The mean age of the study population was 56.52 years (median: 60 years, range: 18–89 years). The study population consisted of 56.3% males (*n* = 722) and 43.7% (*n* = 560) females. The median value of Forced Expiratory Volume in 1 s (FEV1) was 2.60 L (0.35; 6.1) with a median FEV1% of 84% (14.80; 134). The most frequent preoperative comorbidity was hypertension (34.2%). Pre-existing cardiovascular disease was diagnosed preoperatively in 12.1% (*n* = 157 patients) of the study population. A total of 124 out of 1284 patients (9.5%) were active smokers one month before the operation. The comorbidities of the study population are summarized in Table 2.

### 3.3. Postoperative Morbidity and Mortality after Pulmonary Metastasectomy

In 145 patients (11.29%), we identified at least one postoperative complication. The rate of postoperative complications was low. Most complications were minor (9.57%). The most frequent complication was persistent air leaks (*n* = 50 patients, 2.8%). In 1.6% of the study population, major cardiopulmonary complications were diagnosed (*n* = 21 patients).

In-hospital mortality was minimal. Only one patient (*n* = 1) died postoperatively. The 30-day mortality was low. Up to 30 days after PM, eleven patients (*n* = 11) died. Up to 90 days after PM, 38 patients (3.0%) died for any reason, while, for 313 patients (24.4%), the status was unknown. Postoperative complications and mortality are demonstrated in Table 3.

### 3.4. Patient-Associated Risk Factors for Postoperative Morbidity after Pulmonary Metastasectomy

The only patient-associated clinical risk factor for postoperative morbidity was pre-existing cardiovascular comorbidity. Patients with cardiovascular comorbidities before the operation developed complications in 18.4% of cases in comparison with 11.4% of patients without cardiovascular comorbidities (*p* = 0.01). Patients with COPD developed complications in 6.1%. However, here, no statistically significant difference was shown in patients without COPD (*p* = 0.09). In addition, smoking up to 30 days before the operation contributed to an increased rate of postoperative complications (34.0% vs. 25.9%). However, here also, no statistically significant difference was shown (*p* = 0.17). Patient-associated risk factors are analyzed in Table 4.

### 3.5. Surgery-Associated Risk Factors for Patients Undergoing Pulmonary Metastasectomy

Patients undergoing RPM (*p* = 0.01) or major lung resections (*p* < 0.01), as well as open thoracotomy showed statistical significance for increased complications after PM. Open thoracotomy was an adverse factor for morbidity. On the other hand, VATS for PM was protective against postoperative complications (*p* < 0.01). The performance of multiple wedge resection for the complete resection of the lung metastases did not contribute to postoperative complications (*p* = 0.32). Surgical risk factors are demonstrated in Table 5.

### 3.6. Independent Risk Factors for Postoperative Morbidity after Pulmonary Metastasectomy

In the logistic regression analysis, we identified preoperative cardiovascular comorbidities (*p* = 0.01), major lung resections (*p* < 0.01), RPM (*p* = 0.03) and open thoracotomy (*p* = 0.02) as independent factors for postoperative morbidity after PM. Logistic regression analysis is demonstrated in Table 6.

### 3.7. Development of a New Prognostic Score to Predict Postoperative Morbidity after Pulmonary Metastasectomy (Essen Score)

We set a new scoring system to predict postoperative morbidity after PM based on the above-identified independent prognostic factors. We stratified the patients into three distinct groups and one subgroup. The groups were stratified from the lowest to the highest morbidity rate (Essen I–III). The stratification of the score levels according to the independent prognostic factors was carried out empirically.

The first group (Essen I) with the lowest postoperative morbidity included patients undergoing thoracotomy (*n* = 729, complication rate: 14.3%).

The second group (Essen II) included patients that underwent open RPM (*n* = 82). Here, patients undergoing open RPM showed a higher complication rate than patients with open PM (24.4% vs. 14.3%, *p* = 0.006). In addition, we formed a subgroup (Essen IIA), in which patients with preoperative cardiovascular comorbidities undergoing open RPM were included. Here, also, a higher rate of complications was shown (30% vs. 24.4%, *p* = 0.65). However, for the subgroup IIA, no statistical significance was shown.

The third group (Essen III) consisted of patients undergoing open major lung resections. Similarly, here, a higher complication rate was shown compared with open wedge resections (85.7% vs. 12.9%, *p* < 0.01). The results are summarized in Table 7 and Table 8.

In Figure 1, ROC curves of the Essen score are demonstrated.

## 4. Discussion

Pulmonary metastasectomy with curative intent for different primary tumors is a safe surgical procedure with acceptable postoperative morbidity and minimal mortality. We identified, as independent factors for postoperative morbidity, preoperative cardiovascular comorbidities, the performance of open thoracotomy, the need for major lung resection for the complete resection of the lung metastases and repeated pulmonary metastasectomy. We developed a score to predict postoperative morbidity after PM.

PM is associated with low postoperative mortality. Mortality after PM with curative intent ranges between 0% and 2.5% [1,2,3,6,7,8,9]. In our study, in-hospital mortality was minimal. Similar results were shown in the studies of Hassan et al. and Sponholz et al. In both of these studies, the in-hospital mortality was 0%. In addition, after PM, early mortality was shown to be low [9,10]. Hassan et al. showed similar low 30-day mortality, even in the group of elderly in a large series of patients undergoing PM [9]. The 90-day mortality in our study was 3%. Because of these low mortality rates, a further investigation of factors for mortality was not performed.

Our study showed an acceptable rate of postoperative complications. In 11.29% of the study population, at least one complication was diagnosed. However, most of them were minor complications and only 1.6% corresponded to major cardiopulmonary complications. Similar results were shown in the study of Rodríguez-Fuster et al. Here, in patients undergoing PM for colorectal cancer, postoperative morbidity was 15.6% [3]. In our study, except for cardiovascular comorbidities, no other patient-associated factor was identified as an adverse factor for postoperative morbidity. In patients with COPD, postoperative complications were more frequent. However, no statistical significance was shown. Similar results were shown for smokers of up to one month before the operation. COPD is an already-identified adverse prognostic factor for patients undergoing anatomical resections for NSCLC. However, in PM, the role of COPD is controversial [3,9,10]. Rodríguez-Fuster et al. showed that preoperative respiratory comorbidity is an adverse prognostic factor for morbidity [3]. Moreover, Hassan et al. reported COPD to be associated with poor survival for elderly patients after PM [9]. However, these results were not verified in the study of Sponholz et al. [10]. We believe that, in the above-mentioned studies as in our own, a possible selection bias could be present. The COPD patients that underwent PM were in a good clinical condition before the operation and their comorbidities were already treated. Moreover, the resection strategy in PM is the use of lung-sparing wedge resections. As a result, these minor resections acted protectively for these patients.

Cardiovascular comorbidities are an already-identified adverse predictor for postoperative complications. Ambrogi et al. reported cardiovascular comorbidities to be risk factors for patients undergoing lung resections for NSCLC [11]. Sponholz et al. and Rodríguez-Fuster et al. demonstrated similar results for PM [3,10]. Colleagues from the University Clinic of Freiburg showed cardiovascular comorbidities as a risk factor for postoperative minor morbidity and poorer survival after PM. In the same study, the presence of multiple cardiovascular comorbidities per patient and reduced left ventricular function, as well as performing anatomical resections in these patients, were identified as risk factors for postoperative morbidity. Similarly, in both studies, the existence of cardiovascular comorbidities did not lead to increased postoperative mortality [12].

As expected, major pulmonary resections were found to be an independent risk factor for postoperative complications after PM. These findings should be interpreted in the particular context of the surgical strategy for PM. Surgical resections in PM differ largely from resections for primary lung cancer. As mentioned above, the resection of pulmonary metastases is based on lung parenchyma-sparing wedge resections or segmentectomies [1,13,14,15,16,17,18,19,20,21,22,23,24]. In order to achieve the complete resection of multiple lung metastases, the performance of multiple wedge resections is suggested. On the other hand, lung resection for primary lung cancer is based on anatomical lobe resections [2,25]. In our study, wedge resections and segmentectomies consisted of 90.4% of all lung resections. The performance of multiple wedge resections did not influence postoperative morbidity.

We demonstrated open thoracotomy and RPM to be adverse factors for postoperative morbidity. VATS in PM was demonstrated by Rodríguez-Fuster et al. as a protective procedure in PM. VATS has already shown its advantages in decreasing hospital stay length and postoperative pain. In addition, nowadays, with the further evolution of imaging techniques and the preoperative detection of very small lung metastases, their resection could be made possible with VATS and, therefore, minimize the need for thoracotomy [3,26,27,28,29,30]. However, this fact is debated by many surgeons. Many surgeons prefer an extensive palpation of the lung and the resection of all suspicious lesions [2,29]. In the study of Rodríguez-Fuster et al., most resections were performed through thoracotomy [3]. In our study, 56% of cases were operated with open thoracotomy. A selection bias could be present. More complex cases and patients with multiple metastases were operated with an open thoracotomy. On the other hand, single peripheral lung metastases are resected with VATS. Furthermore, concerning RPM as a risk factor for postoperative morbidity, we also believe that this is associated with the complexity of the operation. Hassan et al. showed an increased rate of prolonged air leak and surgical revision after RPM for renal cell carcinoma. However, the fear of complications should not exclude patients from a potential curative RPM. In the study of Hassan et al. and in the current one, the rate of major complications was acceptable [31]. Moreover, it is believed that RPM prolongs survival, as it re-establishes a local control of the tumor disease in the lung and in this way “sets back the clock of the tumor disease” [32]. For this reason, we believe that RPM should be carefully planned and performed as long as local control of the tumor could be achieved.

In our effort to develop a morbidity score, in order to predict postoperative complications after PM, we proposed the new Essen score. To our knowledge, this is the first effort to propose a score that could predict postoperative morbidity after PM. Concerning the resection of lung metastases, Meimarakis et al. and the colleges of the International Registry of Lung Metastases proposed scores that could predict survival after PM. Meimarakis et al. proposed the Munich score for prognosis of survival after PM for renal carcinoma. Their model was based on the complete resection of the lung metastases (R0-resection) [7,33]. Concerning predicting postoperative complications, a significant number of scores have been described from surgeons, anesthesiologists, and intensive care physicians. Many of these scores are based on clinical factors such as age, smoking status and predicted FEV1. For example, in cardiac surgery, the Euroscores are widely used [34]. In thoracic surgery, Amar et al. from the memorial Sloan-Kettering Cancer Center suggested the Clinical Prediction Rule for Pulmonary Complication (CPRPC) [35]. Despite the CPRPC score’s simplicity, Yepes-Temino et al., in another study, described, for their patients, a poor performance of this score [36]. In our case, in stage I, we set the bright basis of patients undergoing thoracotomy. As mentioned above, the need for open surgery for PM could mean a more complex operation. As we added, per stage, more negative prognostic factors for morbidity, the complication rate raised. We believe that the Essen score is suitable for the prediction of postoperative complications after PM. In contrast with other scores that concentrate on clinical patient-associated factors, our score includes prognostic factors that are common concerns in PM and in RPM.

### Limitations

Our study is restricted by some limitations. The current study is firstly restricted by its nature as a retrospective single-center study. In addition, a selection bias could be possible in the patient selection for surgery as well as in the selection of the metastasectomy technique. Patients with multiple lung metastases or in reduced general condition would not be referred to surgery. In addition, for the resection of multiple metastases, an open approach would be preferred. Moreover, the inhomogeneity of the study population should be included in the limitations with the inclusion of patients with so many different primary tumors. Furthermore, because of the retrospective nature of the study and the formation of a very small group of patients, factors such as postoperative mortality, technical complications and the need for blood transfusion by a reoperation or unplanned admission in the ICU could not be investigated. However, we believe that our study with the large patient population undergoing PM is a representative example of the everyday surgical praxis and could be useful in planning surgical resections in this field.

## 5. Conclusions

PM is a safe surgical procedure with acceptable postoperative morbidity and minimal mortality. In patients with cardiovascular comorbidities or requiring an open thoracotomy, major lung resection or RPM, the resection of lung metastases should be carefully planned. In addition, we attempted to validate a new score that would help surgeons to predict and focus on patients with a higher possibility of postoperative complications.

## Figures and Tables

**Figure 1 cancers-15-04355-f001:**
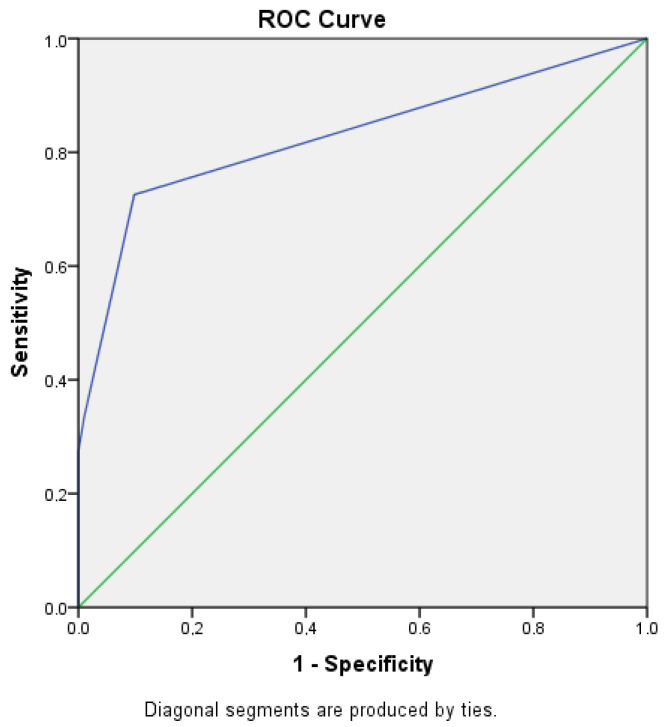
ROC curves of the Essen score.

**Table 1 cancers-15-04355-t001:** Surgical and oncological characteristics of the population.

Oncological Characteristics	
Clinical Factor	*n* Patients (%)
Primary tumor -Bone or soft tissue and other types of sarcoma -Colorectal cancer -Renal cell cancer -Head and neck cancer -Malignant melanoma -Breast cancer -Endometrial and ovarial carcinoma -Germ cell tumors -Urothelial carcinoma -Cholangiocellular carcinoma -Pancreatic cancer -Hepatocellular carcinoma -Gastric cancer -Esophageal cancer -Other	459 (35.3%) 194 (14.9%) 87 (6.7%) 78 (6.0%) 74 (5.7%) 70 (5.4%) 40 (3.1%) 31 (2.4%) 27 (2.1%) 15 (1.2%) 14 (1.1%) 14 (1.1%) 12 (0.9%) 8 (0.6%) 161 (12.4%)
Gender -Male -Female -Unknown	722 (56.3%) 560 (43.7%) 2 (-)
Preoperative therapy -Chemotherapy -Checkpoint inhibitor -Immunotherapy + chemotherapy -Chemotherapy + radiotherapy -Radiotherapy -Unknown	336 (26.3%%) 1 (0.1%) 2 (0.2%) 263 (20.2%) 0 (0.0%) 117 (9%)
Surgical characteristics of the pulmonary metastasectomy	
Surgical approach -Open thoracotomy -VATS	729 (56%) 554 (42.5%)
Surgical resection -Repeated pulmonary metastasectomy	120 (9.2%)
Extent of the lung resection -Bilobectomy -Lobectomy -Pneumonectomy -Segmentectomy -Wedge resection	3 (0.3%) 94 (8.5%) 10 (0.9%) 65 (5.9%) 936 (84.5%)
Lymphadenectomy -Not performed -Lymphadenectomy	802 (61.6%) 482 (37.0%)
Number of wedge resection/operation -Single wedge resection -Multiple wedge resections	419 (45.0%) 513 (55.0%)

VATS: video-assisted thoracic surgery.

**Table 2 cancers-15-04355-t002:** Comorbidities in the study population.

Clinical Characteristics	*n* Patients (%)
Elderly patients (age > 70 years)	261 (20%)
Obesity (BMI > 30)	237 (18.2%)
ASA Score -ASA 3 -ASA 4	438 (33.6%) 34 (2.8%)
Hypertension	445 (34.2%)
Cardiovascular comorbidity	157 (12.1%)
Chronic kidney disease	46 (3.5%)
Gastroesophageal reflux/gastric ulcer	37 (2.8%)
COPD	47 (3.6%)
Active smoker	124 (9.5%)
Insulin-dependent diabetes	31 (2.4%)
Liver disease	49 (3.8%)
Myasthenia gravis	3 (0.2%)
Other comorbidities	589 (45.2%)

BMI: Body-Mass-Index, ASA: American Society of Anesthesiologists, COPD: chronic obstructive pulmonary disease.

**Table 3 cancers-15-04355-t003:** Postoperative morbidity and mortality after pulmonary metastasectomy in the study population.

Postoperative Complication	*n* Patients (%)
At least one complication per patient	145 (11.29%)
Major complications per patient	32 (2.49%)
Major cardiopulmonary complications per patient Prolonged mechanical ventilation with the need for a tracheostomy ARDS Postoperative atelectasis requiring intervention Pneumonia Pleural empyema Bronchopleural fistula Myocardial infarct Postoperative atrial arrythmia Ventricular arrhythmia	21 (1.6%) 1 (0.1%) 1 (0.1%) 5 (0.4%) 4 (0.3%) 3 (0.2%) 1 (0.1%) 1 (0.1%) 6 (0.5%) 3 (0.2%)
Minor complications per patient	123 (9.57%)
Air leak > 5 days	50 (3.8%)
Re-operation for bleeding	10 (0.8%)
Cerebro-vascular complications	2 (0.2%)
Chylothorax	7 (0.5%)
Deep vene thrombosis	1 (0.1%)
O2-need after hospital discharge	2 (0.2%)
Chest wall hematoma	4 (0.3%)
Renal failure	2 (0.2%)
Wound infection	5 (0.4%)
Mortality In-hospital mortality 30-day mortality 90-day mortality	1 (0.07%) 11 (0.856%) 38 (3.0%)

ARDS: Acute Respiratory Distress Syndrome.

**Table 4 cancers-15-04355-t004:** Risk factors for postoperative cοmplications after pulmonary metastasectomy.

Clinical Factor	No Complication Postoperative (*n* Patients, %)	Complication Postoperative (*n* Patients, %)	*p*-Value
Patients with at least one comorbidity preoperatively	1120 (98.8%)	145 (98.6%)	0.89
Elderly patients (age > 70 years)	229 (20.1%)	32 (21.8%)	0.64
Obesity (BMI > 30)	226 (20.3%)	11 (7.6%)	<0.01
ASA score ASA score 3 ASA Score 4	387 (34%) 30 (2.6%)	51 (34.7%) 4 (2.7%)	0.98
ECOG ECOG 2 ECOG 3	29 (2.6%) 8 (0.7%)	4 (2.7%) 1 (0.7%)	0.96
Histology of the primary tumor -Bone or soft tissue and other types of sarcoma -Colorectal cancer	408 (35.9%) 171 (15.0%)	51 (34.7%) 23 (15.6%)	0.47
Hypertension	395 (34.7%)	50 (34.0%)	0.86
Cardiovascular comorbidity	130 (11.4%)	27 (18.4%)	0.01
Chronic kidney disease	39 (3.4%)	7 (4.8%)	0.41
Gastroesophageal reflux/gastric ulcer	33 (2.9%)	4 (2.7%)	0.90
COPD	38 (3.3%)	9 (6.1%)	0.09
Smoking status -Smoker (one month before the operation) -Past smoker	110 (9.7%) 295 (25.9%)	14 (9.5%) 50 (34.0%)	0.17
Insulin-dependent diabetes	28 (2.5%)	3 (2.0%)	0.75
Liver disease	44 (3.9%)	5 (3.4%)	0.78
Myasthenia gravis	2 (0.2%)	1 (0.7%)	0.23
Other preoperative comorbidities	522 (45.9%)	67 (45.6%)	0.93

BMI: Body-Mass-Index, ASA: American Society of Anesthesiologists, ECOG: Eastern Cooperative Oncology Group, COPD: chronic obstructive pulmonary disease.

**Table 5 cancers-15-04355-t005:** Surgery-associated risk factors for patients undergoing pulmonary metastasectomy.

Surgical Characteristics	No Complication Postoperative (*n* Patients, %)	Complication Postoperative (*n* Patients, %)	*p*-Value
Number of PM -Only one PM -RPM	1028 (89.2%) 98 (81.7%)	125 (10.8%) 22 (183%)	0.01
Lymphadenectomy -Not performed -Performed	716 (89.3%) 421 (87.3%)	86 (10.7%) 61 (12.7%)	0.23
Neoadjuvant therapy -Chemotherapy -Immunotherapy -Immunotherapy + chemotherapy -Chemotherapy + radiotherapy	301 (26.6%) 1 (0.1%) 2 (0.2%) 234 (20.7%)	35 (23.8%) 0 (0%) 0 (0%) 29 (19.7%)	0.63
Extent of the resection -Minor resection -Major resection	893 (89.2%) 79 (73.8%)	108 (10.8%) 28 (26.2%)	<0.01
Surgical approach -Open thoracotomy -VATS	625 (85.7%) 511 (92.2%)	104 (14.3%) 43 (7.8%)	<0.01
Number of wedge resections -Single -Multiple	380 (90.7%) 455 (88.7%)	39 (9.3%) 58 (11.3%)	0.32

FEV1: Forced Expiratory Volume in 1 s, PM: pulmonary metastasectomy, RPM: Repeated pulmonary metastasectomy, VATS: Video Assisted Thoracoscopic Surgery.

**Table 6 cancers-15-04355-t006:** Logistic regression analysis of postoperative morbidity factors after pulmonary metastasectomy.

Variables	OR	95% CI	*p*-Value
Preoperative cardiovascular comorbidities	2.299	1.412–3.744	0.01
Major lung resection	2.727	1.678–4.431	<0.01
Repeated pulmonary metastasectomy	1.759	1.040–2.976	0.03
Open Thoracotomy	0.621	0.415–0.930	0.02

**Table 7 cancers-15-04355-t007:** Comparison of postoperative complication rate after pulmonary metastasectomy according to the Essen score.

Essen Score	Patients in the Study Population Group (*n* Patients, %)	Complication Rate (%)	*p*-Value
I (THT)	729 (56%)	14.3%	-
II (THT + RPM)	82 (6.3%)	24.4%	0.006
IIA (THT + RPM + CVC)	10 (0.77%)	30%	0.65
III (THT + maj. resection)	14 (1.09%)	85.7%	<0.01

THT: thoracotomy, RPM: repeated pulmonary metastasectomy, CVC: cardiovascular comorbidities, maj. Resection: major resection.

**Table 8 cancers-15-04355-t008:** Summarizes the OCC, calibration and discrimination of stages of the Essen score.

	OCC	ROC-Analysis
Essen Score	94.6%	0.828 95% CI: 0.795–0.903

OCC: overall correct classification, ROC: receiver operating characteristic curve.

## Data Availability

The data presented in this study are available upon request from the corresponding author.

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
