# Peer review of "New Prognostic Score (Essen Score) to Predict Postoperative Morbidity after Resection of Lung Metastases"

_cancers, 2023, doi:10.3390/cancers15174355_

Round 1

Reviewer 1 Report

From a biostats and clinical epidemiology point of view, this is a well planned original contribute and Essen deserves its score! I have some suggestions for the Authors:

- line 39 and following ones, all the estimations do lack a measure unit! I do suppose OR, don`t it?

- line 42, what about PM anatomical site (you already studied the PM extent) and primary tumor histotype (i.e. comparing neoplasm super-families, like sarcomas vs. g-int cancers), do they really play no role?

- line 107, Fisher` exact test (typo), you may skip chi-square one

- line 122, all continuous covariates have to be reported as median/IQR (no mean/sd)

- line 122, p-values with 3-sign digits everywhere (i.e. line 177)

- table 1, it should be presented as table 2, after table 1 main pts characteristics

- table 1, have you some infos about past smokers too? If any, please add it!

- table 2, rename as table 1, add gender and most common grade/TNM

- table 2, zero pts having had only radiotherapy, can you confirm that!?

- line 153, mortality has to be referred as “up to xx days after PM”, not as a point estimate

- table 6, uni- and multi- variable binary logistic regression model results have to be reported by 3 columns, as OR, 95%CI, p-value; skip Wald test value

- table 6 , I do suppose it`s the multivariable binary logistic regression model? In that case, it`s mandatory to report all the univariable model series too

- line 190, you have to clearly state the rules that drove you to select and fix Essen score levels, actually it`s impossible to derive them from all results – this is a critical point, how have you choosen the covariates cut-offs? Empirically?

- table 8 and figure 1, calculate and report sensitivity and specificity of the model

- line 254, colleagues (typo)

minor

Author Response

From a biostats and clinical epidemiology point of view, this is a well planned original contribute and Essen deserves its score! I have some suggestions for the Authors:

Thank you very much for your corrections and suggestions!

1 line 39 and following ones, all the estimations do lack a measure unit! I do suppose OR, don`t it?

Thank you for your comment. OR is added on the beginning of the parentheses.

2 line 42, what about PM anatomical site (you already studied the PM extent) and primary tumor histotype (i.e. comparing neoplasm super-families, like sarcomas vs. g-int cancers), do they really play no role?

Response: Suggestions were followed. Unfortunately, due to the retrospective nature of our study we could not demonstrate differences between the different anatomical sites. However, we added this lack in the section of limitations.

3 line 107, Fisher` exact test (typo), you may skip chi-square one

Response: correction was performed.

4 line 122, all continuous covariates have to be reported as median/IQR (no mean/sd)

Correction was performed as suggested.

5 line 122, p-values with 3-sign digits everywhere (i.e. line 177)

Corrections were made as suggested

6 table 1, it should be presented as table 2, after table 1 main pts characteristics

Response: correction was performed.

7 table 1, have you some infos about past smokers too? If any, please add it!

Response: Information was added as performed.

8 table 2, rename as table 1, add gender and most common grade/TNM

Response: correction was performed. Unfortunately, due to the retrospective nature of our study and due to the fact that our study is based on an institutional database we could not demonstrate grade and TNM-stage. However, we added this lack in the section of limitations.

9 table 2, zero pts having had only radiotherapy, can you confirm that!?

Response: We confirm this fact. This information was added in the text.

10 line 153, mortality has to be referred as “up to xx days after PM”, not as a point estimate

Response: correction was performed.

11 table 6, uni- and multi- variable binary logistic regression model results have to be reported by 3 columns, as OR, 95%CI, p-value; skip Wald test value

Response: correction was performed.

12 table 6 , I do suppose it`s the multivariable binary logistic regression model? In that case, it`s mandatory to report all the univariable model series too

Thank you very much for your suggestion. Risk factors are also analyzed in Table 5. As our study contains already 8 tables, additional unneeded information would be a strain for the reader.

13 line 190, you have to clearly state the rules that drove you to select and fix Essen score levels, actually it`s impossible to derive them from all results – this is a critical point, how have you choosen the covariates cut-offs? Empirically?

Response: correction was performed.

14 table 8 and figure 1, calculate and report sensitivity and specificity of the model

Thank you very much for the comment. Sensitivity and specificity of the model are reported in Table 8 and figure 1.

15 line 254, colleagues (typo)

Response: correction was performed.

Reviewer 2 Report

Compliments to the Authors. In my opinion the work completely reflects the editorial philosophy of the Journal on the basis of the clinico-pathological evidence in lung pathology (oncology and surgery). It's a significant work because it represents an alternative approach with Essen score (useful tool to predict postoperative morbidity in patients undergoing pulmonary metastasectomy in curative intention), which requires a good integration and collaboration between other medical disciplines.

Author Response

Thank you very much for your comments!

Round 2

Reviewer 1 Report

A "very final" correction for the headers of table 6: confidence intervals are confidence intervals! So, the labels min-max have to be reported as inferior-superior

Author Response

Thank you very much! We performed all suggested corrections.